# Multiparametric Classification of Non-Muscle Invasive Papillary Urothelial Neoplasms: Combining Morphological, Phenotypical, and Molecular Features for Improved Risk Stratification

**DOI:** 10.3390/ijms23158133

**Published:** 2022-07-23

**Authors:** Ivonne A. Montes-Mojarro, Saki Hassas, Sina Staehle, Philip Sander, Niklas Harland, Lina Maria Serna-Higuita, Irina Bonzheim, Hans Bösmüller, Arnulf Stenzl, Falko Fend

**Affiliations:** 1Institute of Pathology and Neuropathology and Comprehensive Cancer Center Tuebingen, Eberhard-Karls-University, 72076 Tuebingen, Germany; ivonne.montes@med.uni-tuebingen.de (I.A.M.-M.); saki.hassas@gmail.com (S.H.); sina.staehle@student.uni-tuebingen.de (S.S.); phillip.sander113@gmail.com (P.S.); irina.bonzheim@med.uni-tuebingen.de (I.B.); hans.boesmueller@med.uni-tuebingen.de (H.B.); 2Department of Urology, Eberhard-Karls-University, 72076 Tuebingen, Germany; niklas.harland@med.uni-tuebingen.de (N.H.); arnulf.stenzl@med.uni-tuebingen.de (A.S.); 3Department of Clinical Epidemiology and Applied Biostatistics, Eberhard Karls University, 72076 Tuebingen, Germany; lina.serna-higuita@med.uni-tuebingen.de

**Keywords:** non-invasive low-grade papillary urothelial carcinoma, high-grade papillary urothelial carcinoma, urothelial neoplasm of low malignant potential, classification

## Abstract

Diagnosis and grading of non-invasive papillary urothelial tumors according to the current WHO classification poses some challenges for pathologists. The diagnostic reproducibility of separating low-grade and high-grade lesions is low, which impacts their clinical management. Whereas papillary urothelial neoplasms with low malignant potential (PUN-LMP) and low-grade papillary non-invasive carcinoma (LG-PUC) are comparable and show frequent local recurrence but rarely metastasize, high-grade papillary non-invasive carcinoma (HG-PUC) has a poor prognosis. The main objective of this work is to develop a multiparametric classification to unambiguously distinguish low-grade and high-grade tumors, considering immunohistochemical stains for p53, FGFR3, CK20, MIB-1, p16, p21 and p-HH3, and pathogenic mutations in *TP53*, *FGFR3*, *TP53*, *ERCC2*, *PIK3CA*, *PTEN* and *STAG2*. We reviewed and analyzed the clinical and histological data of 45 patients with a consensus diagnosis of PUN-LMP (*n* = 8), non-invasive LG-PUC (*n* = 23), and HG-PUC (*n* = 14). The proliferation index and mitotic count assessed with MIB-1 and P-HH3 staining, respectively correlated with grading and clinical behavior. Targeted sequencing confirmed frequent *FGFR3* mutations in non-invasive papillary tumors and identified mutations in *TP53* as high-risk. Cluster analysis of the different immunohistochemical and molecular parameters allowed a clear separation in two different clusters: cluster 1 corresponding to PUN-LMP and LG-PUC (low MIB-1 and mitotic count/*FGFR3* and *STAG2* mutations) and cluster 2, HG-PUC (high MIB-1 and mitosis count/CK20 +++ expression, *FGFR3* WT and *TP53* mutation). Further analysis is required to validate and analyze the reproducibility of these clusters and their biological and clinical implication.

## 1. Introduction

Bladder cancer represents one of the most common cancers in the world with approximately 550,000 cases annually [1]. About 75% of bladder cancers by the time of diagnosis are low-grade, non-muscle-invasive papillary tumors, which are referred to as non-muscle-invasive bladder cancer (NMIBC) [2].

NMIBC includes papillary tumors invading the lamina propria and the connective tissue and are classified as T1, Ta, and Tis, following the TNM classification. According to their morphological features, these tumors can be further classified into low-grade and high-grade neoplasms. Low-grade tumors Ta and T1 frequently recur (31–78%) but infrequently progress to invasion (less than 1% and 45%) in five-year follow-ups, therefore having a good prognosis [3,4,5]. On the other hand, high-grade papillary tumors may progress to muscle-invasive bladder cancer (MIBC), which commonly progresses to metastasis [6]. Therefore, in the management of urothelial carcinoma, determination of the pathological grade aims at stratifying tumors into different prognostic groups to allow evaluation of treatment results and optimize patient management. Several grading systems were proposed for the pathologic grade of non-invasive papillary lesions of the urinary bladder [7]. The former WHO classification from 1973 classified urothelial bladder carcinoma in grades G1, G2 and G3; however, since this has some limitations in defining G2 tumors, it was substituted by the 2004/2016 WHO classification system [8]. This system is widely used today to robustly classify non-muscle invasive urothelial neoplasias among different histologic subgroups, including papillary urothelial neoplasia of low malignant potential referred to as PUN-LMP, low-grade papillary non-invasive urothelial carcinoma (LG-PUC) and high-grade papillary non-invasive carcinoma (HG-PUC). However, this scheme also poses some difficulties in interpretation, resulting in relatively high interobserver variability, ranging from 12–39% in PUN-LMP cases, 21–63% in LG-PUC and 18–56% in HG-PUC [8].

With the advent of next-generation sequencing techniques in recent years, the diagnosis of bladder cancer has focused not only on morphological but also on molecular features. In the UROMOL study, NMIBC tumors were categorized into three classes based on their transcriptional characteristics, which showed differences in the expression of cell cycle genes and differentiation markers [9]. In addition, the Lund University group identified five distinct entities according to activatable targeting and transcription factors, including the Uro A, the genomically unstable, the infiltrative SCC-like and the Uro B subtypes [10], which roughly correspond to the basal and luminal subgroups described in the classification of the University of North Carolina [11]. Those molecular subsets have gained great importance, as they are directly related to clinical prognosis, but have not entered clinical practice yet [11].

The usual treatment for patients with NMBIC is transurethral resection of the bladder tumor (TURBT) with or without intravesical therapy with Bacillus Calmette-Guerin (BCG) to reduce the risk of progression [12]. However, despite the efficacy of BCG treatment, 60% of patients may recur after 12 months and 20% of patients may progress to MIBC [13]. Unsuccessful BCG treatment in patients requires radical cystectomy or chemotherapy and radiation, both of which are associated with considerable morbidity [14]. Therefore, other therapies such as immune checkpoint inhibitors and therapies based on their molecular profile are required to improve the morbidity of bladder carcinoma [15,16].

In this study, we investigated the features of non-invasive PUC at three different levels, including histopathology, immunophenotype and mutational status, aiming for accurate and robust classification. This classification includes the use of morphological features, an immunohistochemical panel including CK20, p16, p21, p53, MIB-1, the mitosis marker p-HH3 and FGFR3 (fibroblast growth factor receptor), and targeted mutation analysis of genes related to recurrence and prognosis, including *FGFR3*, *TP53*, *PTEN*, *PIK3CA*, *ERCC2* and *STAG2*.

## 2. Results

### 2.1. Description of the Cohort

A total of 45 patients with non-invasive urothelial papillary neoplasias classified originally as pTa with good tissue quality, and suitable for further analysis were included in this study. The clinical information is summarized in Table 1. The male-to-female ratio was 2.75:1 and the median age was 75 years (range, 50–95 years). According to the TNM classification, 39/45 (86.6%) patients were classified as pure pTa whereas 6/45 (13.3%) patients had a pT1 lesion at another site. Nineteen cases (42.2%) showed multiple tumor localizations in two or more regions of the bladder, whereas 26 cases showed a solitary lesion. Less than half of the patients recurred (48.4%) and only 52.6% developed no relapse by the two-year follow-up. According to the available clinical data of 44 patients, only 25/44 (56.8%) received immediate therapy, and 13/44 (29.5%) patients received immediate MMC instillations. All patients (12/44, 27.3%) with histological HC-PUC diagnosis were treated with re-TUR-BT and nineteen (19/44, 43.2%) patients were only under medical observance but no treatment was given. Further treatment was given in case of recurrence (Table 1).

### 2.2. Histological Classification

Data of diagnostic concordance among three pathologists blinded to the original diagnosis and the original diagnostic report (ODR) are reported in Table 2. The interobserver variability in the diagnosis of PUN-LMP was modest (57%), good (67%) in cases of LG-PUC and excellent (87%) in cases of HG-PUC. The most significant interobserver variability was found when comparing PUN-LMP and LG-PUC diagnosis, with a rather low Fleiss Kappa index of 0.43 (*p* = <0.001) with a 95% confidence interval between 0.398–0.635. A consensus diagnosis was established when an agreement was met by at least two independent observers. According to the consensus, only 18% of the cases were classified as PUN-LMP, but half (51.1%) of the patients fell into the category of LG-PUC and 33.3% were classified as HG-PUC. In the follow-up, a higher recurrence rate was observed in cases of HG-PUC, as expected. Cases of PUN-LMP demonstrated a higher median free relapse (34 months) in comparison to 23 months and 5 months of LG and HG-PUC, respectively. Two-year recurrence analysis revealed a higher recurrence rate (79.5%) for HG-PUC, in comparison to LG-PUC (42.7%) and PUN-LMP (33.3%) (*p* = 0.14, Figure 1). The baseline clinical characteristics and follow-up were described according to the consensus diagnosis in Appendix A.

### 2.3. Immunohistochemical Characteristics and Correlation with Tumor Grade

Spearman’s rank correlation matrix of the histoscores rendered no positive correlation between the diagnostic categories except for the proliferation index, which was positively correlated with the mitotic count assessed by the p-HH3 staining (Figure 2). CK20, p53 and the proliferation rate (MIB-1) are suitable tools to identify cases with higher histological grade and adverse outcomes. These three markers showed higher histoscores in cases of LG-PUC and HG-PUC versus PUN-LMP cases (*p* < 0.05) (Table 3 and Figure 2). The tumor suppressor p53 displayed heterogeneous staining among the three subgroups but displayed higher expression in cases classified as LG-PUC and HG-PUC in comparison to PUN-LMP (*p* = 0.06). Additionally, cases classified as PUN-LMP showed weaker staining for CK20, p53 and a lower proliferation and mitosis rate evaluated by p-HH3 in comparison to the LG- or HG-PUC cases (*p* < 0.001). LG-PUC cases demonstrated similar histoscores for CK20 and p53 compared to the HG-PUC group. In contrast, the proliferation rate (MIB-1) and mitosis count showed higher scores in HG-PUC cases. Some cases of PUN-LMP and LG-PUC showed higher mitotic counts, but in these cases, mitosis was restricted to the basal layer, in contrast to HG-PUC. The cell cycle proteins p21 and p16 were heterogeneously stained among the cases and did not show a correlation with the histological grade. In total, 28/37 (75.7%) cases show a lack of FGFR3 staining from all cases but six cases (6/20, 30%) classified as LG-PUC displayed overexpression of FGFR3. In general, only 16/25 (64%) cases show a lack of FGFR3 expression correlated with its mutational status since some mutations may result in a non-functional protein.

### 2.4. Mutational Landscape

In total, 37 cases were analyzed using targeted NGS. The mean average read depth of sequencing was 6820 reads (1141–23,628). The distribution of mutations is depicted in Figure 3 and detailed information is summarized in Appendix A. In total, 80 mutations were identified, 66/80 missense point mutations (82.5%), nine nonsense mutations (11.25%), and five frameshift mutations (6.25%). Analysis by SIFT, PolyPhen-2 and CADD predicted a damaging effect for all the reported missense mutations. Variant allele frequencies (VAF) of the mutations in all genes ranged from 3% to 93% (mean 34%).

Independent of the grading, *FGFR3* was the most frequently affected gene in this series. Cases classified as PUN-LMP showed mainly *FGFR3* alterations (57%, 4/7) and an absence of *TP53* mutations. In addition, mutations detected in *ERCC2, PTEN* and *PIK3CA* were mostly concurrent and overall infrequent (28.5%, 2/7). Low-grade PUC cases likewise displayed frequent *FGFR3* mutations in 17/20 (85%) cases and 11 cases showed concurrent alterations in *PIK3CA* (63.6%, 7/11), *STAG2* (54.5%, 6/11) and *ERCC2* (45.4%, 5/11). *TP53* mutations were present in 2/20 (10%) cases. HG-PUC cases showed a lower frequency of *FGFR3* alterations (40%, 4/10) and more *TP53* alterations (30%, 3/10). *TP53*-mutated cases always revealed a high *TP53* histoscore, but conversely not all cases with high histoscores exhibited a *TP53* genetic alteration. *PTEN* mutations were rare but present in both PUN-LMP and HG-PUC cases. Of all genes investigated, only *TP53* mutations were correlated significantly with relapse-free survival (*p* = 0.024, Appendix A).

### 2.5. Cluster Analysis Integrating Morphology, Immunophenotype and Mutational Status

Cluster analysis clearly divided non-invasive papillary urothelial tumors into two different subgroups. Cluster 1 represents a mixture of PUN-LMP and LG-PUC cases, displaying a low proliferation ratio and mitotic count, together with frequent *FGFR3* and STAG2 mutations, whereas cluster 2 contains HG-PUC with high proliferation index and mitotic count, strong CK20 expression, *TP53* mutation but *FGFR3* WT (Table 4). Cluster analysis correlated strongly with the clinical behavior and showed lower relapse-free survival in group 2, (cases classified as HG-PUC) in comparison to group 1, (cases classified as PUN-LMP and LG-PUC), (*p* = 0.029, Figure 3B,C). Additional cluster analysis was performed using three different subgroups to determine whether PUN-LMP and LG-PUC cases could be segregated. (Figure 3B, left side). However, a mixture of cases of PUN-LMP and LG-PUC was found in groups 1 and 2 and the cluster did not correlate with histological grading. Almost all cases classified as HG-PUC were gathered into group 3 (Figure 3B,C, left side).

The Kaplan–Meier analysis showed a strong correlation with the clusters, as recurrence-free survival rates were consistently higher in groups 1 and 2, with overlap between LG-PUC and PUN-LMP (*p* = 0.055). Group 3 consistently showed a higher recurrence rate (Figure 3C, left panel). A proportional Cox regression model was performed to estimate the association between RFS and the clusters analysis. Univariate and multivariate analyses confirmed that clinical characteristics and treatment did not influence RFS. Therapy by RE-TURB denoted a significant correlation with recurrence, but it was not considered since RE-TURB was exclusively performed in cases of HG-PUC (Appendix A).

Multivariable analysis confirmed that clinical characteristics and treatment did not influence relapse-free survival and that relapses were independently associated with the clusters groups resulting from histological characteristics, protein expression and mutational status (Appendix A).

## 3. Discussion

In this study, we employed a multiparametric classification of non-invasive papillary urothelial tumors, encompassing mutational status, immunophenotype and histopathology to try to improve risk assessment over histological grading alone. In previous as well as our current analysis, poor to moderate interobserver reproducibility was noted for grading of papillary urothelial tumors. However, a precise grading of the tumor is essential for therapy decisions and to decrease the risk of recurrence and invasion [2,17]. The differences in grading may be due to different weights given to the morphological features (architecture, cytology, mitoses, etc.) for grading, despite numerous attempts to clearly define the grading criteria [18]. However, it is known that grading systems always involve a degree of subjectivity that affects interobserver reproducibility. In papillary urothelial tumors, the most difficult challenge is the distinction of PUN-LMP from LG-PUC, since both entities have the typical papillary architecture, i.e., papillae covered by urothelium lacking apical mitotic activity, with the only defining features being the presence and amount of atypia and disorder of the epithelium, which may be assessed subjectively. In cases of PUN-LMP, the urothelium appears thicker but atypia should be virtually excluded, whereas in the cases of LG-PUC the papillae display an orderly appearance at low magnification and atypia is present but mild [19,20]. In our study, in agreement with the literature, PUN-LMPs showed the lowest degree of interobserver agreement and reproducibility and an increasing trend to diagnose LG-PUC instead of PUN-LMP. In addition, differences in recurrence rates were relatively similar when comparing PUN-LMP and LG-PUC. The recurrence rate for PUN-LMP cases varies from study to study in the literature, probably an outcome likely influenced by the criteria used for classification. Recurrences range from 17 to 35% for PUN-LMP, 48–71% for LG-PUC and up to 78% for HG-PUC cases. In contrast, the risk for invasion is 1–3% for PUN-LMP, 5% for LG-PUC and up to 45% for HG-PUC cases [21].

Since histological features demonstrate obvious limitations, other tools such as mitosis count, proliferation rate and expression of specific immunohistochemical markers were studied as ancillary methods for grading urothelial neoplasms. Studies using mitotic activity evaluated by semi-quantitative methods showed low reproducibility for grading and poor correlation with prognosis [18]. However, studies in breast cancer suggest that mitotic counting can be improved by quantitative methods [22]. In our study, mitotic counting was performed using p-HH3 immunostaining to reduce subjectivity, although it has to be mentioned that p-HH3 staining cannot be compared directly to mitosis counting on H&E sections. A high mitotic index (>20 mitoses in 10 HPF) was observed in all cases of HG-PUC demonstrating that mitosis count with p-HH3 is a useful parameter for grading. The proliferation rate assessed by MIB-1 staining is also a helpful feature, but the differences in proliferation rate between PUN-LMP and LG-PUC are subtle and overlap, precluding its use as a sole parameter for grading [23]. Some authors reported CK 20 and p53 staining as objective markers for urothelial dysplasia [24]. CK20 is usually only expressed in umbrella cells in normal urothelium, whereas diffuse full thickness CK20 staining is characteristic of HG-PUC [25]. However, discordant CK20 expression was reported in cases of PUN-LMP and LG-PUC; therefore, this parameter should not be used in isolation [26].

In recent years, the advent of robust high throughput sequencing technologies has resulted in significant progress in our understanding of the mutational landscape of urothelial carcinoma and the prognostic impact of specific alterations, leading to several molecular classifications of bladder cancer. NMIBC are reported as genomically stable tumors characterized by *FGFR3* mutations or chromosomal translocations (in around 70% of low-grade tumors) [27,28], which may contribute to early urothelial hyperplasia [29]. In concordance with these data, *FGFR3* mutations were the most common genetic alteration in our series (64.1% of the cases). As expected, *FGFR3* mutations were associated with a non-high-grade phenotype. Activating mutations in *PIK3CA* and inactivating mutations in the cohesin complex tumor suppressor gene *STAG2* seem to play a secondary role in the pathogenesis since most of these mutations co-occur with *FGFR3* alterations [30,31,32], as observed in the majority of the cases in this study. This suggests that *PIK3CA* and *STAG2* alterations themselves are related to a low risk of progression [33,34]. Conversely, high-grade papillary carcinoma and invasive tumors frequently display alterations in the *TP53*, *ERCC2* and *PTEN* genes and pathways [35,36]. Pathogenic *TP53* mutations were rarely found, but when present, they were exclusively identified in LG- and HG-PUC representing, as in other cancers, a genetic signature of malignancy [37,38]. The DNA helicase *ERCC2*, previously described as a biomarker for sensitivity to cisplatin therapy in MIBC, similarly appears to play a role in bladder cancer oncogenesis [39,40]. However, as observed in this series, it also can be identified in all categories of NMIBC. The tumor suppressor *PTEN* is mostly genetically inactivated in MIBC (90%) in comparison to NMIBC (40%) [41]. In this series, however, inactivated *PTEN* was also found in PUN-LMP cases (2/4 cases mutated). It remains to be seen whether rare cases of PUN-LMP and LG-PUC with mutations in *PTEN* or *TP53*, generally associated with high-grade tumors, show a higher progression risk.

The joint evaluation of immunohistochemical and molecular markers in our principal component analysis allowed us to identify two groups with significantly different progression risks, despite the small number of cases. In addition, this cluster analysis using multiple parameters suggests that PUN-LMP and LG-PUC are closely related, likely representing a spectrum of biological behavior. Given that the same clinical and therapeutic approach is used for PUN-LMP and LG-PUC, the rationale for their distinction, which currently is based on histological features with considerable subjectivity in their interpretation, is unclear and requires further analysis with larger numbers of cases [19,21,42,43]. Multiparametric analysis, which employs additional protein and genetic markers, can be useful in NMIBC grading and might improve the prediction of clinical behavior and risk of recurrence and progression.

The results of this study should be interpreted with caution, as this study represents only a limited number of patients in whom only a limited number of protein markers and mutations were analyzed. This work primarily denotes a proof of principle study in which the main objective is to show that a multiparametric approach holds some promising results, which should be tested in a larger cohort. In the era of personalized medicine, molecular panels detecting mutations or their immunohistochemical surrogates are critical for selecting the best therapy and predicting the prognosis of patients with bladder cancers. Since PUN-LMP and UPC represent early diagnoses in the clinical course of bladder cancer patients, their multiparametric analysis including protein biomarkers and mutational profiling holds promise and should be performed in larger cohorts.

## 4. Materials and Methods

### 4.1. Tumor Samples

In this study, we included all samples from the Institute of Pathology and Neuropathology, University Hospital Tuebingen with a diagnosis of non-invasive papillary urothelial tumor between 2013 and 2020 meeting our selection criteria: biopsies from patients with papillary non-muscle invasive bladder cancer graded as PUN-LMP, LG-PUC or HG-PUC. Representative blocks were selected for analysis. In addition, nineteen samples of bladder biopsies without tumor or dysplasia served as negative controls. This study was approved by the Ethics Committee of the Medical Faculty of the University of Tuebingen (547/2021BO2).

### 4.2. Histological Classification According to the 2016 WHO Classification

For reclassification according to the WHO 2016 classification, 3 µm thick sections stained with hematoxylin and eosin (HE) were blindly evaluated by three pathologists. Cases were categorized as PUN-LMP, non-invasive LG-PUC or HG-PUC. The evaluation of cytological and architectural features was performed at low, medium and high magnification (100×, 200× and 400×) [8,44]. PUN-LMP was diagnosed in cases with discrete papillae, slender and not fused, covered by mostly normal urothelial cells with minimal or absent cytologic atypia, mitosis, and necrosis. The non-invasive LG-PUC biopsies showed orderly arranged papillae with mild cytological atypia and rare mitosis in the lower half of the urothelium. The non-invasive HG-PUC cases demonstrated a completely disorderly appearance at low magnification due to architectural disorganization, moderate to high-grade atypia and mitotic activity throughout the height of the epithelium [44].

### 4.3. Clinical Data

The medical records were reviewed including age, sex, disease presentation, location, extent, TNM staging, treatment and follow-up. Recurrence-free survival (RFS) was calculated from the date of diagnosis to 5-year or the last follow-up.

### 4.4. Immunohistochemistry

Immunostaining was performed using an automated stainer (Ventana Medical Systems, Tucson, AZ, USA), according to the manufacturer’s protocol. The electrocharged slides were stained with CK20 (M7019, Dako Cytomation, Glostrup, Denmark), p16 (06594441001, CINtech, Roche, Manheim, Germany) p21 (556431, BD Pharmigen, San Jose, CA, USA), p53 (DO-7, Novocastra, Leica Biosystems, Wetzlar, Germany), MIB-1 (M7240, Dako Cytomation, Glostrup, Denmark), p-HH3 (117C826, IMGENEX, San Diego, CA, USA) and FGFR3 (sc-13121, Santa Cruz Biotechnology, Heidelberg, Germany). Immunohistochemistry was quantified using the histoscore, as described below. The staining was scored according to both intensity and the cell percentage. Intensity was assessed in the membrane and cytoplasm and scored as 0: negative, +: weak, ++: moderate, +++: strong. Histoscore was then calculated by multiplying the intensity of the stains by the percentage of positively stained cells. In addition, p-HH3 immunohistochemistry was used to identify cells undergoing mitosis. Staining was assessed by counting the number of stained cells in 2 mm^2^, approximately 10 consecutive high-power fields [HPF] of upper layers of the urothelium with a Zeiss Axioskop microscope (Zeiss, Oberkochen, Deutschland).

### 4.5. DNA Extraction

Genomic DNA was extracted from remaining biopsy material using the Maxwell^®^ RSC DNA FFPE Kit and the Maxwell^®^ RSC Instrument (Promega, Madison, WI, USA), according to the manufacturer’s instructions. DNA was quantified with the Qubit Fluorometer employing the Qubit dsDNA HS Assay Kit (Thermo Fisher Scientific, Waltham, MA, USA), according to the manufacturer’s protocol. Quality control polymerase chain reaction (PCR) was performed to identify the amplifiable DNA length as previously reported [45]. Cases with at least 200 bp amplifiable DNA were selected for NGS targeted analysis.

### 4.6. Targeted Next Generation Sequencing

Targeted mutation analysis was performed by Next Generation Sequencing (NGS) (Ion GeneStudio S5 prime, Thermo Fisher Scientific, Waltham, MA, USA) using an AmpliSeq Custom Panel covering the most common recurrently mutated genes in papillary non-muscle invasive urothelial bladder cancer (complete coding sequence: *ERCC2*, *FGFR3*, *PIK3CA*, *PTEN*, *STAG2* and *TP53* (Appendix A). Amplicon library preparation and semiconductor sequencing was carried out according to the manufacturers’ manuals using the Ion AmpliSeq Library Kit v2.0, the Ion Library TaqMan Quantitation Kit, the Ion 510, Ion 520 and Ion 530 Kit—Chef and the Ion 520 Chip Kit (Thermo Fisher Scientific, Waltham, MA, USA). Five paraffin-embedded cases of papillary non-muscle invasive urothelial bladder served to validate the NGS panel. Variant calling of non-synonymous somatic variants compared to the human reference sequence was performed using Ion Reporter Software (Thermo Fisher Scientific, Waltham, MA, USA).

### 4.7. Cluster Analysis

Clustering analysis was conducted to identify groups of patients with similarities. Analysis was performed in only 37 cases because data were missing in eight patients. Standardization of variables (Versi) was performed to obtain a mean of zero and standard deviation of one. The clustering analysis was based on the following variants: histology classification (PUN-LMP, LG-PUC and HG-PUC), immunohistochemistry (CK20, p53, MIB-1, p16, p21, FGFR3 and p-HH-3) and mutational status of *ERCC2*, *FGFR3*, *PIK3CA*, *PTEN*, *STAG2*, *TP53* (mutated vs. nonmutated). The Partitioning around Medoids (PAM) algorithm of the K-Medoids cluster was used to group patients based on their similarity since this algorithm is less sensitive to noise and outliers. In this method, each cluster is represented by one of the data points in the cluster. These points are named cluster medoids (object within a cluster for which average dissimilarity between it and all the other members of the cluster are minimal). [46,47]. The calculation of the distance between two real-valued vectors was performed using the Euclidean distance, which is defined as follows: (1)$$d_{\mathrm{euc}}(x,\;y)=\sqrt{\sum_{i=1}^{n}(xi-yi{)}^{2}}$$

The average silhouette method was applied to estimate the optimal number of clusters.

### 4.8. Statistical Analysis

Statistical analysis was performed using R Software Version 4.0 (R Foundation forStatistical Computing, Vienna, Austria). Categorical variables were described using frequencies and proportions; numerical variables were reported as either means and standard deviation (±SD) or medians and interquartile range (IQR), depending on the distribution of the data. Normality of the distribution was assessed by investigating kurtosis, skewness as well as QQ plots. Categorical variables were assessed by using Chi-square test. Fleiss’ Kappa tests were performed to measure interobserver variability. Relapse-free survival (RFS) was analyzed by Kaplan–Meier method and log-rank test was used to test differences between groups. The follow-up was defined as the period from 24 months until the occurrence of the outcome, or otherwise censored. Univariable and multivariable Cox proportional hazard regression were used to assess the association between cluster and relapse-free survival. Candidate risk factors for the multivariate model were selected based on the clinical considerations and the statistically significant results from the bivariate analyses. Backward selection was performed to sequentially remove variables from the model. Multicollinearity was checked by matrix correlation. Results were reported as hazard ratio and 95% confidence interval. The proportion hazard assumption was evaluated using the Schoenfeld residuals. All statistical tests were two-sided, and *p* < 0.05 was considered to indicate statistical significance.

## 5. Conclusions

Over the last decades, classifications combining different levels of complexity in cancer diagnosis have emerged as an important tool for tumor detection and stratification. They provide a strong correlation with clinical behavior and offer biological insight and understanding of neoplasms which show difficulties in reproducible classification such as PUN-LMP. In this work, we confirm that several histologic, immunophenotypic and genetic features show overlap between PUN-LMP and LG-PUC, also reflected in the difficulties in reliably separating these two entities. This ultimately leads to the question of whether PUN-LMP and LG-PUC represent a biological continuum, which is captured insufficiently by histological grading and requires the addition of molecular and phenotypical features as ancillary tools for better risk stratification. However, due to the limited number of cases and genes investigated, further studies are required to validate and extend these results.

## Figures and Tables

**Figure 1 ijms-23-08133-f001:**
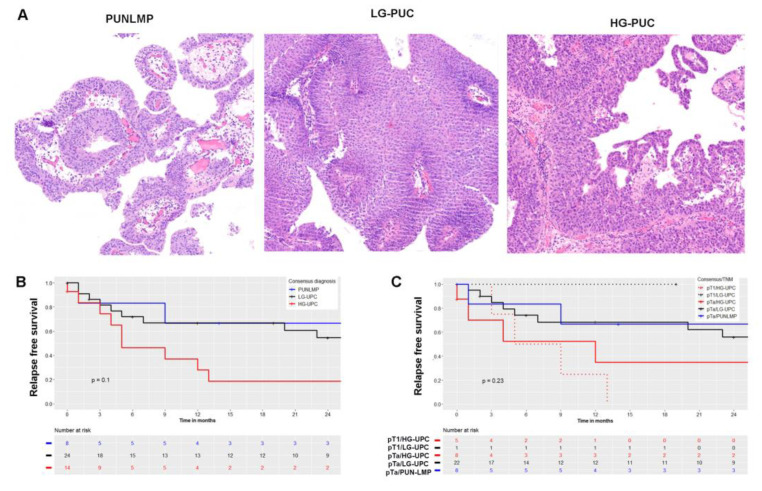
Histological grading and relapse-free survival. (**A**). Representative histology images displaying the main features of different tumor grades (PUN-LMP, LG-PUC and HG-PUC). (**B**). Kaplan–Meier analysis of relapse-free survival comparing tumor grades by consensus diagnosis. (**C**). Kaplan–Meier analysis of relapse-free survival comparing TNM classification and tumor grades. PUN-LMP, papillary urothelial neoplasia of low malignant potential; LG-PUC, low-grade urothelial papillary carcinoma; HG-PUC, high-grade urothelial papillary carcinoma.

**Figure 2 ijms-23-08133-f002:**
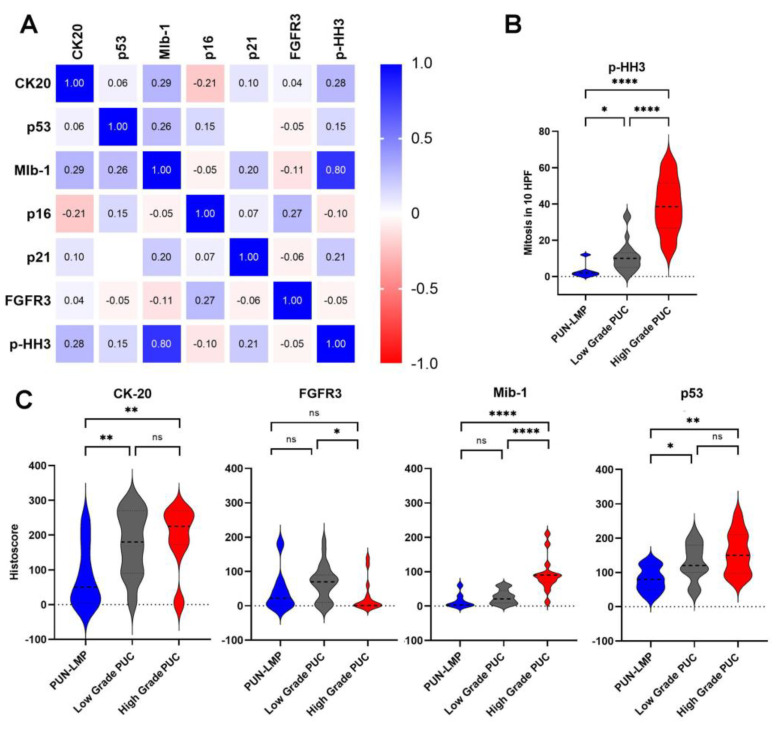
Immunohistochemical features. (**A**). Spearman’s rank correlation matrix and correlation significances of immunohistochemical variables. Values show the Spearman rank results; blue colors show positive correlations and red colors negative correlations. (**B**). Violin plots comparing the number of mitotic counts in 10 high power fields (HPF) stained with p-HH3 among non-invasive PUC categories. (**C**). Violin plots of the different histoscores of the immunohistochemistry markers, denoting the non-invasive PUC entities. For statistical analysis unpaired *t*-test was used, ns= not significant, * *p* < 0.05, ** *p* < 0.01, **** *p* < 0.0001.

**Figure 3 ijms-23-08133-f003:**
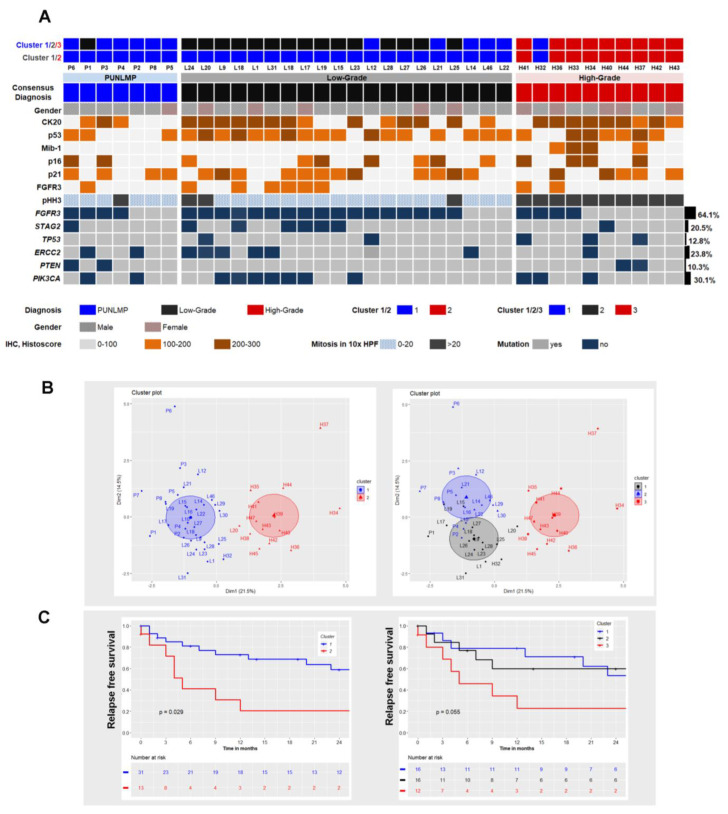
(**A**). Overview of the mutational and immunohistochemical results per diagnostic category. (**B**). Cluster analysis of two and three subgroups (right and left, respectively). (**C**). Kaplan–Meier plots of relapse-free survival using the cluster analysis displayed in cluster plots.

**Table 1 ijms-23-08133-t001:** Clinical characteristics of cases with non-invasive urothelial papillary neoplasia.

	*n* = 45 (%)
Median age (range)	75 years (50–95)
Gender	*n* (%)
Male	33 (73.3)
Female	12 (26.6)
M:F Ratio	2.75:1
Tumor Stage, TNM	*n* (%)
pTa	39 (86.6)
pTa and pT1	6 (13.3)
Localization of tumor	
2 or more locations	19 (42.2)
Lateral walls	10 (22.2)
Posterior wall	6 (13.3)
Trigonum	3 (6.6)
Neck/Apex	3 (6.6)
Dome of the bladder	2 (4.4)
Anterior wall	2 (4.4)
Detrusor muscle in the histological slides	30 (66.7)
Immediate therapy after diagnosis * (*n* = 44)	*n* (%)
No therapy	19 (43.2)
Immediate intravesical instillation of MMC	12 (27.3)
Re-TUR-BT	13 (29.5)
Recurrence in two-year follow-up % *	48.4%
Further therapy during the clinical course	
No further therapy	12 (27.3)
Further BCG or MMC instillations	10 (22.7)
TUR-B	31 (70.5)
Cystectomy and TUR-B:	2 (4.5)
Cystectomy, TUR-B and BCG therapy	2 (4.5)

MMC: Mitomycin C; BCG, Bacillus Calmette-Guerin (BCG); TUR-BT, transurethral resection of bladder tumor. * Percentages are calculated for 44 patients since some clinical data were not available for one case.

**Table 2 ijms-23-08133-t002:** Interobserver variability among the histological categories of the WHO 2004/2016.

Grade	ODR	Diagnosis P 1	Diagnosis P 2	Diagnosis P 3	Consensus Diagnosis	Interobserver Variability (%)	Fleiss Kappa
PUN-LMP	18 (40%)	16 (35.6%)	8 (17.8%)	6 (13.3%)	8 (17.8%)	57%	0.43 **
LG-PUC	14 (31.1%)	15 (33.3%)	22 (48.9%)	27 (60%)	23 (51.1%)	67%	0.41 **
HG-PUC	13 (28.9%)	14 (31.1%)	15 (33.3%)	12 (26.7%)	14 (31.1%)	87%	0.81 **

PUN-LMP, papillary urothelial neoplasia of low malignant potential; LG-PUC, low-grade urothelial papillary carcinoma; HG-PUC, high-grade urothelial papillary carcinoma; P 1, 2, 3, pathologist 1, 2 and 3. ** *p* value < 0.001 Fleiss Kappa.

**Table 3 ijms-23-08133-t003:** Histoscores of cases according to the histological grading.

	CK20	Mib-1	p-HH3 *	FGFR3	p16	p21	p53
PUN-LMP	60 (6–180)	3 (3–30)	2 (1–12)	9 (1–40)	10 (15–270)	150 (60–170)	90 (50–130)
LG-PUC	202 (110–270)	25 (9–45)	10 (5–15)	70 (10–82.5)	85 (17.5–180)	110 (57.5–160)	127.5 (100–182.5)
HG-PUC	225 (172–270)	90 (57–97.5)	38.5 (25.3–51.5)	26.3 (10–82.5)	25 (3–225)	97.5 (6–191.3)	142.5 (97.5–210)
*p*-value	0.027	<0.001	<0.001	0.032	0.065	<0.001	0.066

Histoscore median (p25–p75); PUN-LMP, Papillary Urothelial Neoplasia of low malignant potential; LG-PUC, low-grade urothelial papillary carcinoma; HG-PUC, high-grade urothelial papillary carcinoma. * Number of mitoses in 10 high power fields (HPF).

**Table 4 ijms-23-08133-t004:** Comparison of protein and mutational status among the clusters.

Marker	Cluster 1	Cluster 2	*p*-Value
Immunohistochemistry	median (p25–p75)	median (p25–p75)	Wilcoxon rank sum test
CK20	160 (80–240)	225 (172.5–270.0)	0.150
p53	110 (80–160)	158.7 (107.5–212.5)	0.013
MIB-1	15 (3–33.9)	90 (68.5–97.5)	<0.001
FGFR3	60 (1–80)	3 (1.0–65.0)	0.101
p-HH3 *	8 (3–12)	38.5 (25.7–51.5)	<0.001
Mutational status	*n* (%)	*n* (%)	χ^2^ test
*FGFR3*	18 (40.0%)	3 (6.0%)	0.028
*STAG2*	8 (17.2%)	1 (2.2%	0.236
*TP53*	2 (4.4%)	5 (11.1%)	0.023
*ERCC2*	7 (15.5%)	4 (8.8%)	0.717
*PIK3CA*	10 (22.2%)	3 (6.6%)	0.724

Immunohistochemistry: data are expressed as median (p25–p75) of histoscore values or (*) number of mitoses in 10 HPF. Mutational status is represented by number of mutated cases (%).

## Data Availability

Not applicable.

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
