# Peer review of "Multiparametric Classification of Non-Muscle Invasive Papillary Urothelial Neoplasms: Combining Morphological, Phenotypical, and Molecular Features for Improved Risk Stratification"

_ijms, 2022, doi:10.3390/ijms23158133_

Round 1

Reviewer 1 Report

The authors have submitted a manuscript entitled "Multiparametric classification of non-muscle invasive papillary urothelial neoplasms: combining morphological, phenotypical, and molecular features for improved risk stratification". 

It is an interesting study addressing an important problem. However, some issues should be addressed:

The number of patients (n=45) is low. When divided into subgroups (PUN-LMP (n=8), non-invasive LG-PUC (n=23), and HG-PUC (n=13), the statistical power is not high enough.

Unfortunately, there is not enough information on the therapy of the patients. How many patients received single immediate instillations after the TUR-B? How many patients did receive a follow-up instillation therapy and how was it performed? Did this influence the recurrence rate?

If the groups are heterogeneous concerning the therapy it is hard to draw conclusions.

How was the TUR-B performed (piecemeal or en-bloc-resection)? Was detrusor muscle present in all specimens ensuring the quality of resection? 

Which patients did receive a second resection and did this influence the recurrence rate?

I recommend to do a multivariate analysis to check the influence of all these parameters.

The manuscript should be re-arranged: the Material and Methods section should precede the Results and Discussion section.

Author Response

Answer to Reviewer 1.

We thank the Reviewer for the time taken to read our manuscript, for the criticisms and recommendations to improve this manuscript.

We recognize that one of the limitations of the study is the restricted number of patients. However, this study only represents a proof of principle study to show that multiparametric approach by using histology, protein expression and molecular analysis are important to understand the biology of the tumors and may improve prognostication.

In order, to address the concerns about the impact of the therapy (BCG instillations, type of TURB) we collected the therapy data missing an performed an extra statistical analysis. Changes in the manuscript are highlighted in red. Please see below:

“Material and methods 

 Relapse free survival (RFS) was analyzed by the Kaplan Meier method and long rank test was used to test differences between groups. The follow-up was defined as the period from 24 months until the occurrence of the outcome, or otherwise censored. Univariable and multivariable Cox proportional hazard regression were used to assess the association between cluster and relapse free survival. Candidate risk factors for the multivariate model were selected based on the clinical considerations and the statistically significant results from the bivariate analyses. Backward selection was performed to sequentially remove variables from the model. Multicollinearity was checked by matrix correlation. Results were reported as hazard ratio and 95% confidence interval. The proportion hazard assumption was evaluated using the Schoenfeld residuals. All statistical tests were two-sided, and p<0.05 was considered to indicate statistical significance”

“Results

According to the available clinical data of 44 patients, only 25/44 (56.8%) received immediate therapy, 13/44 (29.5%) patients received immediate MMC instillations.  All patients (12/44, 27.3%) with histological certain of HC-PUC diagnosis were treated with re-TUR-BT and nineteen (19/44, 43.2%) patients were only under medical observance but not treatment was given. Further treatment was given in case of recurrence (Table 1).

A proportional Cox regression model was performed to estimate the association between RFS and the clusters analysis. Univariate and multivariate analyses confirmed that clinical characteristics and treatment did not influence RFS. Therapy by RE-TURB denoted a significant correlation with recurrence, but it was no considered since RE-TURB was exclusively performed in cases of HG-PUC (Supplementary Table 4). Multivariable analysis confirmed that clinical characteristics and treatment did not influence relapse-free survival and that relapses were independently associated with the clusters groups resulting from histological characteristics, protein expression and mutational status (Supplementary tables 5 and 6).”

Reviewer 2 Report

Dear Editor, thank you so much for inviting me to revise this manuscript about urothelial carcinomas.

This study addresses a current topic.

The manuscript is quite well written and organized. English could be improved.

Figures and tables are comprehensive and clear.

The introduction explains in a clear and coherent manner the background of this study.

We suggest the following modifications:

·      Introduction section: although the authors correctly included important papers in this setting, we believe the authors should better discuss the treatment scenario for urothelial carcinomas and some recently published studies should be cited within the introduction ( PMID: 33516645;      PMID: 32498352 ), only for a matter of consistency. We think it might be useful to introduce the topic of this interesting study.

·      Methods and Statistical Analysis: nothing to add.

·      Discussion section: Very interesting and timely discussion. Of note, the authors should expand the Discussion section, including a more personal perspective to reflect on. For example, they could answer the following questions – in order to facilitate the understanding of this complex topic to readers: what potential does this study hold? What are the knowledge gaps and how do researchers tackle them? How do you see this area unfolding in the next 5 years? We think it would be extremely interesting for the readers.

However, we think the authors should be acknowledged for their work. In fact, they correctly addressed an important topic, the methods sound good and their discussion is well balanced.

One additional little flaw: the authors could better explain the limitations of their work, in the last part of the Discussion.

We believe this article is suitable for publication in the journal although major revisions are needed. The main strengths of this paper are that it addresses an interesting and very timely question and provides a clear answer, with some limitations.

We suggest a linguistic revision and the addition of some references for a matter of consistency. Moreover, the authors should better clarify some points.

Author Response

We thank the Reviewer for the time taken to read our manuscript, for the criticisms and the suggestions in order to improve this manuscript.

  • To address the comments of the reviewer in the introduction section. We added a paragraph, which is highlighted in red about the treatment of NMIBC and we also annexed the suggested references (references in paper 15 and 16).  Changes in the manuscript are highlighted in red. Please see below.

“The usual treatment for patients with NMBIC is transurethral resection of the bladder tumor (TURBT) with or without intravesical therapy with Bacillus Calmette-Guerin (BCG) to reduce the risk of progression.1. However, despite the efficacy of BCG treatment, 60% of patients may recur after 12 months and 20% of patients may progress to MIBC 2 Unsuccessful BCG treatment in patients require radical cystectomy or chemotherapy and radiation, both of which are associated with considerable morbidity3. Therefore, other therapies such as immune checkpoint inhibitors and therapies based on their molecular profile are required to improve the morbidity of bladder carcinoma4, 5

  • We also expanded the discussion to address the questions brought up by the reviewer concerning the potential of the work, the confrontation of the knowledge gaps and the future of the research.The limitations of the study were also discussed as requested by the reviewer. Please see below.

“The results of this study should be interpreted with caution, as this study represents only a limited number of patients in whom only a limited number of protein markers and mutations were analyzed. This work denotes primarily a proof of principle study in which the main objective is to show that a multiparametric approach holds some promising results, which should be tested in larger cohort. In the era of the personalized medicine, molecular panels detecting mutations or their immunohistochemical surrogates are critical to select the best therapy and predict the prognosis of patients with bladder cancers. Since PUN-LMP and UPC represent early diagnoses in the clinical course of bladder cancer patients, their. multiparametric analysis including protein biomarkers and mutational profiling holds promise and should be performed in larger cohorts. ”

Additional references

  1. Hall MC, Chang SS, Dalbagni G, Pruthi RS, Seigne JD, Skinner EC, et al. Guideline for the management of nonmuscle invasive bladder cancer (stages Ta, T1, and Tis): 2007 update. The Journal of urology 2007 Dec; 178(6): 2314-2330.10.1016/j.juro.2007.09.003:doi:

  1. van den Bosch S, Alfred Witjes J. Long-term cancer-specific survival in patients with high-risk, non-muscle-invasive bladder cancer and tumour progression: a systematic review. Eur Urol 2011 Sep; 60(3): 493-500.10.1016/j.eururo.2011.05.045:doi:

  1. Shore ND, Palou Redorta J, Robert G, Hutson TE, Cesari R, Hariharan S, et al. Non-muscle-invasive bladder cancer: An overview of potential new treatment options. Urol Oncol 2021 Oct; 39(10): 642-663.10.1016/j.urolonc.2021.05.015:doi:

  1. Mollica V, Rizzo A, Montironi R, Cheng L, Giunchi F, Schiavina R, et al. Current Strategies and Novel Therapeutic Approaches for Metastatic Urothelial Carcinoma. Cancers (Basel) 2020 Jun 2; 12(6).10.3390/cancers12061449:doi:

  1. Rizzo A, Mollica V, Massari F. Expression of Programmed Cell Death Ligand 1 as a Predictive Biomarker in Metastatic Urothelial Carcinoma Patients Treated with First-line Immune Checkpoint Inhibitors Versus Chemotherapy: A Systematic Review and Meta-analysis. Eur Urol Focus 2022 Jan; 8(1): 152-159.10.1016/j.euf.2021.01.003:doi:

Round 2

Reviewer 1 Report

The authors picked up many suggestions. The added information revealed th limitations of this study. Unfortunately, many patients included were not treated accordung to the guidelines.

Nevertheless, the paper could be accepted for publication as the authors added a critical statement: "The results of this study should be interpreted with caution, as this study represents only a limited number of patients in whom only a limited number of protein markers and mutations were analyzed. This work denotes primarily a proof of principle study in which
the main objective is to show that a multiparametric approach holds some promising re
sults, which should be tested in larger cohort."

The Materials and Methods Section is still positioned after the Discussion section. This should be changed.

Author Response

We thank the referee for reviewing our manuscript again.

As requested by the reviewer, the "Material and Methods" section has been placed before the "Results" section.  This change was highlighted in red.

Reviewer 2 Report

Acceptance.

Author Response

We would like to thank the reviewer for accepting our work for publication in the International Journal of Molecular Sciences.